# Comparison between a Calving Predictive System and a Routine Prepartal Examination in German Holstein Heifers and Cows

**DOI:** 10.3390/vetsci9040192

**Published:** 2022-04-15

**Authors:** Lara Górriz-Martín, Annabel Koenig, Klaus Jung, Wiebke Bergforth, Dirk von Soosten, Martina Hoedemaker, Árpád Csaba Bajcsy

**Affiliations:** 1Clinic for Cattle, University of Veterinary Medicine, 30173 Hannover, Germany; lara.gorriz.martin@tiho-hannover.de (L.G.-M.); annabel.koenig@tiho-hannover.de (A.K.); martina.hoedemaker@tiho-hannover.de (M.H.); 2Institute for Animal Breeding and Genetics, University of Veterinary Medicine Hannover, 30559 Hannover, Germany; klaus.jung@tiho-hannover.de; 3Institute for Food Quality and Food Safety, University of Veterinary Medicine Hannover, 30559 Hannover, Germany; wiebke.bergforth@tiho-hannover.de; 4Institute of Animal Nutrition, Friedrich-Loeffler-Institute (FLI), Federal Research Institute for Animal Health, 38116 Braunschweig, Germany; dirk.von_soosten@fli.de

**Keywords:** calving prediction, Moocall, dairy cattle

## Abstract

The objective was to validate the efficacy of Moocall^®^ comparing it to a routine clinical examination. Altogether 38 Holstein cows were enrolled in this study (Moocall^®^ group: 16 heifers and 8 cows; control group: 9 heifers and 5 cows). Clinical examinations were performed every 6 h over the 7 days period before the predicted calving date. The examined traits were changes in pelvic ligament relaxation, edema of the vulva, teat filling, vaginal secretion, tail tip flexibility, tail raising and behavior. There were no significant differences in Moocall^®^ alerts between heifers and cows. The time lag between the first warning of Moocall^®^ and the onset of labor was 21.2 ± 20.2 h (max: 95.4 h; min: 0.1 h; *p* = 0.87) for heifers and 29.6 ± 29.6 h (max: 177.8 h; min: 0 h; *p* = 0.97) for cows. Linear models including Moocall^®^ alerts showed a significantly better fit to the time until calving than models without Moocall^®^ information (without variable selection: *p* = 0.030, with variable selection: *p* < 0.01). In the best-fitting model, class 2 alerts (enhanced tail activity over 2 h) contributed with a higher significance (*p* < 0.01). Vice versa, models including additional traits were outperformed the use of Moocall^®^ alerts alone. In the best fitting model, class 2 alerts (enhanced tail activity during 2 h) contributed with a higher significance (*p* < 0.01) than any of the best clinical predictive parameters, such as pelvic ligament relaxation (*p* = 0.01), tail tip flexibility (*p* = 0.01) or behavior (*p* = 0.01).

## 1. Introduction

Over the last decades, the expansion of the size of dairy farms has led to new challenges in the management of cattle [1]. Automated systems have proven to be suitable tools in milk production and herd management [2,3]. However, there are still some key aspects related to animal health and welfare that require more efficient smart devices to detect potentially threatened animals. A crucial time in the productive life of dairy cattle remains the events surrounding calving and the periparturient days. Over the last years, several procedures and devices have been developed to predict calving [4]. Despite technical development, the most sensitive predictive parameter still remains blood (serum or plasma) concentration of progesterone (P4; sensitivity: 97.8%), which drops dramatically before parturition; however, its specificity and positive predictive value are lower (73.9% and 79.7%, respectively). Furthermore, its exact measurement is time-consuming if performed in a laboratory [4,5]. Further parameters that can be evaluated to predict imminent labor are based on clinical signs (pelvic ligament relaxation, body temperature decrease, teat filling and udder skin distension) and behavior (lying transitions, feed and water intake, and rumination), as well as a combination of both. Examination of the pelvic ligament relaxation together with teat filling 12 h prior to parturition yielded higher values of sensitivity and negative predictive values and rather similar values of positive predictive values in cows (94.7%, 99.3%, and 14.9%, respectively) but lower values of specificity (57.8%) than in heifers [6]. Results in which calving predictions were based on an observed decrease in feeding time and rumination activity of the animals in the period prior to parturition are irregular in literature, due to the significant individual variations and the low number of animals used in such trials [4,7,8,9]. One parameter that has gained attention during the last years which could be effective in the prediction of calving is tail movement. It was observed that approximately 5 days before calving occurs, cows raise their tail for defecation and urination more frequently and longer than during the previous days [10]. Moreover, Wehrend et al. [11] stated that tail raising occurs from the first stage of labor onwards, independently of urination or defecation. Additionally, several studies have demonstrated that dairy cows and heifers significantly kept their tail risen 2–6 h before calving [12,13,14]. Another study using a tail-mounted inclinometer sensor (Moocall^®^) stated a notable variability in the sensitivity (19 to 75%) and specificity (from 63 to 96%), depending on the interval of clinical examinations preceding the onset of parturition (1, 2, 4, 12, and 24 h until calving). This study also described events with a swollen or painful tail [15]. Other birth monitoring system were tested in primiparous dairy heifers. These detected the onset of stage II of labor earlier than conventional monitoring executed by farm staff. Due to deficits in the device, the authors advise the improvement of the sensor before its use in primiparous heifers [16].

Moocall^®^ (Moocall, Moocall Ltd., Dublin, Ireland) is a device based on an accelerometer system detecting tail raising and overactivity. Placed at the tail 4 days prior to the predicted calving date (PCD), the Moocall sensor produces alarms in form of short message services (SMS) in case of overactivity.

Therefore, the aims of this study were to investigate the efficiency of Moocall in predicting labor in dairy cows and heifers and to compare calving detection through Moocall with a routine clinical examination at a dairy cattle farm.

## 2. Materials and Methods

### 2.1. Animals

In total, 38 late pregnant German Holstein cattle (25 heifers and 13 cows) from the dairy herd of the Institute of Animal Nutrition, Friedrich-Loeffler-Institute (FLI), Braunschweig, Germany were used in the present study. Animals were randomly assigned either to the Moocall (*n* = 24) or to the control group (*n* = 14). Cows were housed in free stall barns and milked twice daily until dried off. They were fed once daily, in the morning with a total mixed ration (TMR). Heifers were kept in a separate barn and on pasture until they reached approximately 240 days of pregnancy, when they were relocated to the calving barn. This study was conducted between October and December 2017.

### 2.2. Calving Management

A list of heifers and cows was obtained monthly, based on PCD using on-farm computer records (Agrocom Superkuh; CLAAS KGaA mbH, Harsewinkel, Germany). Pregnant cows (with at least one previous lactation) were dried off approximately 60 days before PCD and moved into the dry pens immediately after the last milking. Similarly, pregnant heifers were moved into the dry pens (separated from dry cows) approximately 42 days before PCD. All heifers and cows were housed in similar prepartum pens and moved into adjacent individual maternity pens for calving with straw bedding. The animals were moved into the calving pens after first signs of udder filling. The diets of the animals were formulated to meet the nutritional requirements of dry dairy cows stated by the German Society of Nutrition Physiology [17]. Pregnant animals were monitored by on-farm personnel for imminent signs of parturition (presence of the fetal membranes or claws of the calf outside the rima vulvae). Independent of the on-farm personnel, one trained veterinarian performed an examination of the animals every 6 h. The examined traits were changes in pelvic ligament relaxation, edema of the vulva, teat filling, vaginal secretion, flexibility of the tail tip, tail raising, and behavior. The calving ease of cows (assistance provided at birth) were recorded using a four-point scale (1 = no assistance provided; 2 = light assistance provided by one person without mechanical traction; 3 = mechanical extraction of the calf with an obstetric calf-puller; and 4 = severe dystocia: surgery or fetotomy needed). Additionally, date and time of calving were recorded. After calving, cows were processed (e.g., harvesting of colostrum) and moved to the postpartum pens. The trained veterinarian did not influence the obstetrical decisions taken by the on-farm personnel.

### 2.3. Clinical Examination

Clinical examinations including the traits changes in pelvic ligament relaxation, edema of the vulva, teat filling, vaginal secretion, tail tip flexibility, tail raising and behavior were carried out every 6 h during 7 days before PCD in all animals. All these parameters were assessed manually by a trained veterinarian. The variations in the parameters of pelvic ligament relaxation and edema of the vulva were graded as mild, moderate, and severe. Additionally, the pelvic ligament was considered to have concave conformation if the ligaments were extremely relaxed. The variables of teat filling, vaginal secretion, tail tip flexibility, and tail raising were considered dichotomous. Behavior changes were considered when the animal was observed moving restlessly through the calving pen, licking, or sniffing the ground. The onset of calving was set by the appearance of the allantochorion membranes or one or two extremities at the vulva.

### 2.4. Moocall

Altogether, five Moocall sensors were used. The device was manually attached to the base of the tail four days prior to PCD, calculated as 280 days after artificial insemination (AI) or if it was suspected that calving was approaching/would begin earlier than 276 days post AI (Figure 1). Moocall delivered two types of signals depending on the tail position changes in case of an imminent calving: SMS type 1 alert (SMS1), if enhanced activity was registered over one hour and SMS type 2 alert, (SMS2) if high activity continued in the consecutive hour. In the Moocall-group, clinical examinations were performed and documented, independently of the alerts emitted by Moocall.

### 2.5. Statistical Analysis

Date distribution was tested for normality by means of the Shapiro-Wilk test. Normally distributed variables were compared with the *t*-test and expressed as mean ± standard deviation (SD). Non-normally distributed variables were compared with the Wilcoxon rank sum test and expressed as median ± median absolute deviation (MAD). Correlations between the duration from AI to birth and the duration from AI to first Moocall SMS, second Moocall SMS, respectively, were calculated using Pearson’s correlation coefficient R. Three linear regression models (calving prediction without Moocall information, with Moocall information, and with the best clinical parameter predictors) were additionally fitted using the duration from AI to birth as dependent variable. Either clinical parameters alone or clinical parameters together with Moocall-SMS information generated by Moocall activities were used as explanatory variables. In both cases, automatic backward variable selection was used to identify optimal models with regard to Akaike’s Information Criterion (AIC) to avoid overfitting. Furthermore, residual plots were generated to illustrate deviations between true and predicted birth time. Deviations larger or smaller than 6 h or 12 h were determined as percentage of studied individuals. In addition, coefficients of determination R^2^ were used to judge the model fit. Statistical analyses were performed using the software SAS 9.4.1. (SAS Institute Inc., Cary, NC, USA) and R 3.5 (www.r-project.org, accessed on 30 March 2022).

## 3. Results

### 3.1. Gestation Length, Calving Ease and Calf Vitality

#### 3.1.1. Moocall Group

Gestation length was 276.6 ± 9.1 days in heifers (*n* = 16) and 283.9 ± 4.2 days in cows (*n* = 8) (*p* < 0.05; mean ± SD). Gestation was shorter than 280 days in 10 out of 15 heifers (66.7%, one AI data is missing) and in 2 out of 8 cows (25%). Ten animals (41.7%) calved within less than 4 days after Moocall had been mounted, in accordance to the installation criteria provided by the manufacturer (Moocall Users guide). Two animals (8.3%) calved on the day when Moocall was installed. Nine animals (37.5%) needed more time (9.1 ± 2.5 days) to give birth than estimated with the PCD. Three animals calved 4.6 ± 3.8 days before Moocall had been installed following the criterion to wait with its installation until 4 days prior to PCD. These animals were instrumented with Moocall because of the outcomes of the clinical examination. Calving monitoring in 6 h intervals gave a number of 18.4 ± 8.1 times in heifers and 33.2 ± 19.9 times (mean ± SD) in cows. Spontaneous calvings occurred in 17 cases. One cow had mild and 6 heifers had moderate (*n* = 3) or severe (*n* = 3) difficulties at calving and needed various calving assistance (grade 2 or 3). Caesarean sections and fetotomies were not needed in any of the cases. While 22 vital calves were born, one was weak and another one died at birth; both of these were born from heifers.

#### 3.1.2. Control Group

Gestation length was 276.5 ± 4.1 days in heifers and 279.2 ± 6.7 days in cows, which did not show a statistically significant difference (mean ± SD) and was shorter than 280 days in 7 out of the 9 heifers (77.8%) and in 3 out of 5 cows (60%). Calving monitoring sessions in 6 h intervals was 10.5 ± 3 (median ± MAD) in four heifers. Three of the heifers involved in this part of the study calved 24 h within the calculated time for the beginning of the clinical examinations (PCD—7 days), while in two cases the performance of the calving monitoring was not possible because these heifers were not allocated in the calving pen. The number of calving monitoring sessions averaged 24.2 ± 26.2 in cows and sessions were performed in all the five cows. Spontaneous calving was observed in five heifers, while assistance was needed in four heifers, from which one case needed only mild, and two others severe, assistance. In the other case, the calf needed mild assistance and two others required severe assistance due to its posterior presentation. Seven calves from heifers were vital, one was weak at birth, and in one further case data were not available. Four calves from cows were vital and another one was dead at calving.

### 3.2. Findings of the Calving Monitoring 24 h Prior to Calving

Clinical signs that changed within the final 24 h prepartum are summarized in Table 1.

### 3.3. Moocall Activity

The five Moocall sensors produced altogether 258 SMS. 154 of them informed about elevated activity, 21 informed about the fact that Moocall was detached from the tail, and 5 warned that it was time to temporarily remove the device from the tail because the time of attachment had exceeded 4 days. The rest of the SMS (58) reported about battery level, activation, deactivation, or remounting events or about testing the SMS function.

Moocall sent 6.1 ± 3.9 SMS-s (max: 14; min: 2) in heifers and 6.5 ± 4.3 SMS-s (max: 13; min: 2) in cows about elevated activity. In heifers, 62 SMS1 and 40 SMS2 were sent. In cows, 35 SMS were class 1 SMS and 17 class 2 SMS. There were no statistically significant differences in Moocall activity between heifers and cows (SMS1: *p* = 0.87; SMS2: *p* = 0.97; Table 2).

The coefficients of determination between the true duration from AI to birth and the duration from insemination to SMS1 were R^2^ = 0.73 for cows and R^2^ = 0.91 for heifers. The coefficients of determination between the true duration from AI to birth and the duration from insemination to SMS2 were R^2^ = 0.76 for cows and R^2^ = 0.94 for heifers (Figure 2). The proportion of animals with deviations of +/− 6 h between true and predicted birth ranged between 34.7% (SMS1) and 52.2% (SMS2), and between 39.1% (SMS1) and 56.5% (SMS2) for deviations of +/− 12 h (Table 3). Considerable proportions of deviations between 1 and 8 days were observed for SMS1 and SMS2 (Figure 2).

### 3.4. Statistical Models with and without Involving Moocall Information

Models without Moocall-SMS information yielded worse-fitting results than models including Moocall-SMS information (Figure 3). Specifically, adjusted R² was higher for models with Moocall-SMS information. Furthermore, AIC was generally lower (i.e., better) after automatic variable selection. Using likelihood ratio tests, the full model with SMS information (model 3 shown in Figure 3) showed a significantly better fit to the full model without SMS information (*p* = 0.03) but not a better fit than model 2, i.e., model 1 after automatic variable selection (*p* = 0.16). However, model 4, i.e., with SMS information and variable selection, showed a significantly better fit to model 2 (*p* < 0.01). Therefore, and with regard to the coefficient of determination R² and the AIC, we can argue that model 4 fits best to the time until birth. Particularly, the factor SMS2 showed the most significant effect among all predictors in model 4 (*p* < 0.01), i.e., better than the best clinical predictive parameters pelvic ligament relaxation (*p* = 0.01), tail tip flexibility (*p* = 0.01), or behavior (*p* = 0.01).

Studying the deviations between true and predicted birth times proportions of deviations +/*−* 6 h or of +/*−* 12 h were smaller as when using SMS1 or SMS2 alone (Table 3). However, the largest deviations observed with the four linear models were only up to +/− 48 h which is a clearly better performance than when using SMS1 or SMS2 alone (Figure 4).

### 3.5. Remarks When Using Moocall

Ten out of the 24 (41.6%) animals kept the device permanently attached. The mean number of device losses was 1.4 ± 1.9 (max: 6, min: 0) for heifers and 1.0 ± 1.9 (max: 4, min: 0) for cows. In 18 (75%) of these cases the device was detached or lost by the animals, or it was necessary to remove it temporarily because of reaching the recommended 4 days, which is the permitted duration threshold for wearing the device (*n* = 2; 8.3%) or due to tail swelling (*n* = 3; 12.5%). Moocall reported about the loss of the device consistently in one case, often in 3 cases, and in 4 cases it did not inform at all.

## 4. Discussion

Accurate prediction of the onset of calving remains a challenge in the management of cattle, independent of intention (beef or dairy) and size of the herd [18]. Continuous supervision demands time and requires experience to identify parturient animals and to recognize if professional obstetric assistance is needed. Physiological changes in approaching calving have been exhaustively described in the literature [6,7,14,18,19]. Along with the accepted knowledge, the parameters we found to predict parturition more accurately were relaxation of the pelvic ligaments, teat filling, and flexibility of the tail tip. Tail raising was not among the best predictors in our study, but it is a common finding in the literature [11,13]. This inconsistency was probably because the clinical supervisions in the present study occurred punctually at a certain time and not carried out using long-time intervals. In those moments, presumably due to indirect interactions between the animal and the observer based on the unavoidable visual contact, despite the very gently performed regular checkings, the tail movement occurred with a reduced frequency. Indeed, only 5 out of 38 animals (13.2%) were documented to present tail raising 24 h before parturition. Interestingly, two of those animals were heifers, allocated in the Moocall group. We do not preclude the possibility that Moocall may have influenced this observation, however, the fact that 22.2% of the heifers in the control group also showed tail raising makes it improbable. The first statistical model (Figure 3) yielded the highest R-value (R = 0.97) for cows, which means that the examination of the clinical signs including pelvic ligament relaxation, edema of the vulva, teat filling, vaginal secretion, tail tip flexibility, tail raising, and behavior in 6-h intervals suffices for an accurate calving prediction in high pregnant multiparous cows. Nonetheless, in heifers this was not the case (R = 0.92). We observed higher R-values in heifers when the best predictive parameters were selected (pelvic ligament relaxation, teat filling, and tip tail flexibility; R = 0.96). This suggests that the evaluation of edema of the vulva, vaginal secretion, tail raising, and behavior do not improve the prediction in primiparous animals and their changes during the last days before parturition should be interpreted cautiously. Streyl et al. [6] reported similar results. In this study, pelvic ligament relaxation combined with teat filling yielded the best values for forecasting either the presence of an initiating labor or its absence within 12 h. Furthermore, authors found that clinical signs they had monitored were more precise for predicting calving in cows than in heifers.

The use of the smart device Moocall as an assisting tool to help in the management of parturition in high pregnant cattle could be weighed but we consider it essential to highlight some critical points when using it in a herd.

The first one is related to its different efficacy when cows and heifers are compared. Moocall generated lower R-values in cows than in heifers (SMS1: cows: R = 0.73; heifers: R = 0.91. SMS2: cows: R = 0.76; heifers: R = 0.94; Figure 2). This is probably due to an enhanced tail raising in heifers before calving. This difference in tail raising between cows and heifers has already been reported by Miedema [13], who stated that heifers showed tail raising 4 h and cows 2 h before calving. A further aspect needing revision is the point of time at which the device has to be installed on the tail. The manufacturer advises to install the device when the cow is close to calving. In the present study, 4 days prior to PCD were considered reasonable. At this farm the PCD value was calculated from the average 280 days pregnancy length. However, 57% of our animals showed a gestation length shorter than 280 days, which agrees with the data of Grunert (1993; [20]). The effect of this correction in the utilization of Moocall should be tested in future studies.

One further aspect is how the rate of successfully implemented Moocall sensors could be increased. In our study, only less than half of the animals (41.6%) kept the device permanently attached. In the rest, the device was accidentally detached or had to be removed because of swelling of the tail or surpassing the 4 days time being mounted, according to manufacturers’ recommendation. Detachment of the device was not always reported by the devices. These observations have already been reported [15]. Another important point was the extremely extended time lapse between SMS1 or SMS2 and the onset of calving and the high variability between the SMS and the onset of calving in some cases. In this respect, no differences between cows and heifers were found (Table 2). It should be mentioned that it is important to consider the possibility that when Moocall is implemented and warns with an SMS1 for the first time, although the onset of calving may occur during the next hour, it may also take place 95.4 h thereafter in heifers or even 177.8 h thereafter in cows, which is longer than a week. Nonetheless, if the whole dataset is considered, we observed that models without Moocall information represented by SMS yielded worse fitting results than models where SMS were included (Figure 3). Additionally, AIC was consistently better in models where Moocall was included. The lowest (i.e., better) AIC value was achieved with SMS2. To our knowledge this is the first time that the efficacy of Moocall or an electronic sensor based on tail movements and rise in cattle was investigated with linear models and subsequent automatic backward variable selection to identify optimal models about Akaike’s Information Criterion. Therefore, a comparison with other studies is not possible and we may infer that SMS2 seems to inform more precisely about calving than SMS1 and the clinical parameters.

In the case that Moocall is going to be used to help in the calving management of a herd, we strongly recommend to specify the measures that have to be taken in case of a warning. We observed cases where Moocall sent up to 14 SMS to inform about an imminent calving, which thereafter in some cases took more than 95 h (SMS1) before calving had occurred. It has to be asked, which actions should be taken by the responsible person in charge in such cases? Should the affected animal be examined after each warning, thereby obviously disturbing the animal and her approaching calving process? Or should it only be gently observed from a safe distance with the risk of missing key aspects of the labor? Should stipulated measures also be performed during the night, even if Moocall warns with SMS1? One can never know if the last warning will be indeed the last. This has two consequences: (1) “no calving” cannot be ruled out from the moment a warning is generated in form of SMS, and (2) prediction of calving within a short time seems unsure or is impossible. If one cannot rule out that the device is lost and Moocall did not inform about the detachment of the device, to wait for the first SMS2 without examining the animals should be considered hazardous. These facts are more crucial if several late pregnant animals in a herd are provided with Moocall at the same time.

## 5. Conclusions

In summary, the SMS2 warnings sent by the smart device Moocall are more precise at detecting labor. Furthermore, a combined model with SMS2 alerts and additional traits as predictors such as pelvic ligament relaxation, tail tip flexibility, or behavior performed better than SMS alerts alone or than models without SMS alerts. Moocall may therefore serve as an additional support for calving management, since it can inform one about a possible calving onset in an inaccurate moment of the consecutive time. Among other limitations, this device did not predict “no calving” or calving within a clearly defined time frame. For these reasons, other alternatives should be used; in most cases, performing clinical examination and observing the signs of the animals, including pelvic ligament relaxation, teat filling, and tail tip flexibility [6]. These clinical parameters have demonstrated once again to be sufficient for the prediction of calving in both cows and heifers. However, this information is gathered through direct contact with each animal individually, meaning the resulting labor can be intensive. Therefore, although Moocall could help in the efficacy of detecting labor, it cannot replace the more reliable, regularly performed, thorough clinical examinations.

## Figures and Tables

**Figure 1 vetsci-09-00192-f001:**
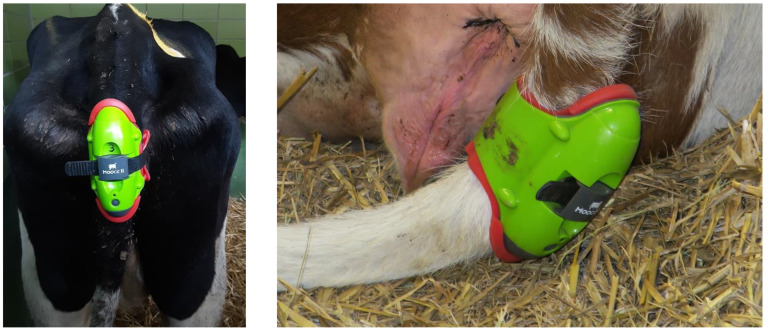
The Moocall^®^ (Moocall Ltd., Dublin, Ireland) remote calving sensor attached to the tail of late pregnant cows.

**Figure 2 vetsci-09-00192-f002:**
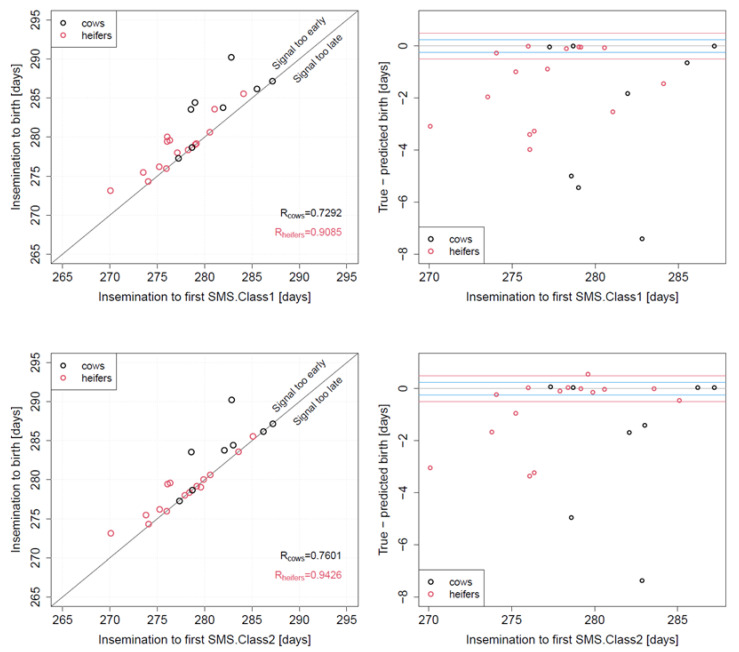
(**Left**): correlation between duration from insemination to SMS1 (**top**; enhanced tail activity registered over one hour) and SMS2 (**bottom**; high tail activity continued during the consecutive hour) emitted by the remote calving sensor Moocall attached to the tail of late pregnant animals, and the true duration from insemination to birth. (**Right**): difference between true and predicted birth date using SMS1 (**top**) or SMS2 (**bottom**).

**Figure 3 vetsci-09-00192-f003:**
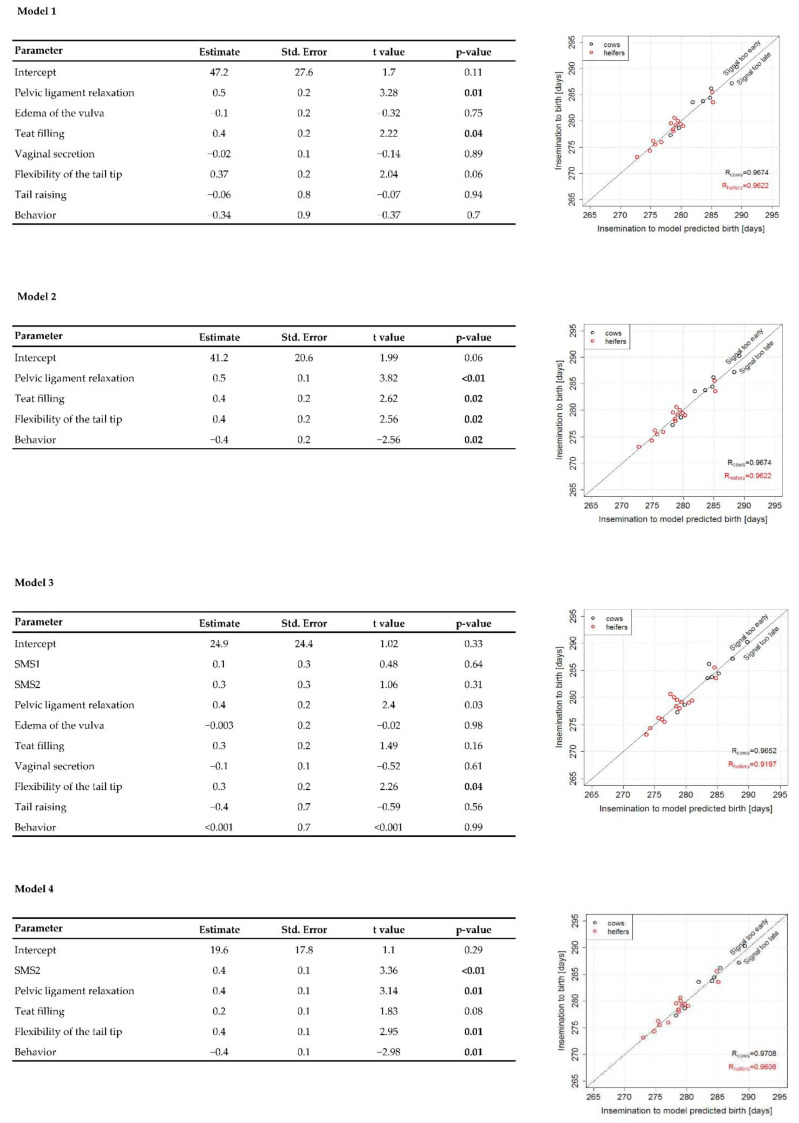
Statistical models without (models 1 and 2) and with (models 3 and 4) information from the remote calving sensor Moocall in form of short message services (SMS) to predict calving without (models 1 and 3) and with (models 2 and 4) automatic parameter selection. Models 1 and 3 are full models, and models 2 and 4 are models after automatic variable selection. SMS1: enhanced tail activity registered over one hour; SMS2: high tail activity continued during the consecutive hour.

**Figure 4 vetsci-09-00192-f004:**
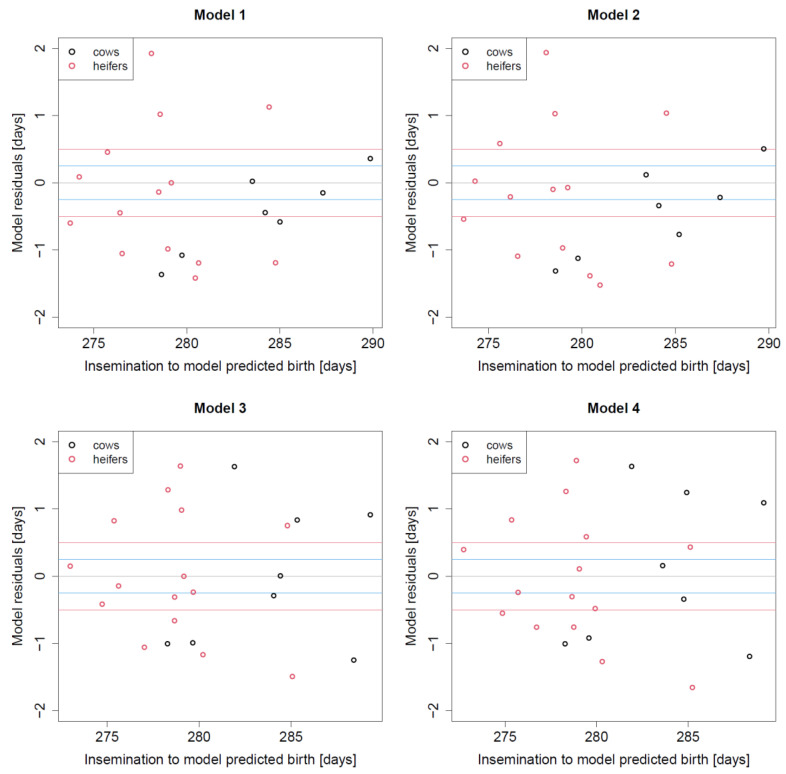
Residual plots for models 1–4, showing model residuals versus predicted birth date. Blue lines indicate deviations +/− 6 h, red lines indicate deviations +/− 12 h.

**Table 1 vetsci-09-00192-t001:** Number of animals (n, percentage (%)) sorted by age (cows vs. heifers) and experimental group (animals instrumented with the remote calving sensor Moocall^®^ (Moocall group) vs. Control group) that conveyed changes in pelvic ligament relaxation, teat filling, vulva edema, vaginal secretion, flexibility of the tail tip tail raising, and changes in behavior 24 h prior to calving.

Experimental Group	Animals	Pelvic Ligament Relaxation			Teat Filling	Vulva Edema			Vaginal Secretion	Flexibility of the Tail Tip	Tail Raising	Changes in Behavior
**MooCall-group**		**mild to moderate**	**moderate to severe**	**severe to concave**		**absent to mild**	**mild to moderate**	**moderate to severe**				
	Heifers *n* = 16	2 (12.5%)	6 (37.5%)	2 (12.5%)	3 (18.8%) **	1 (6.25%)	3 (18.8%)	2 (12.5%) *	7 (43.8%)	4 (25%)	2 (12.5%)	
	Cows *n* = 8	1 (12.5%)	2 (25%)	3 (18.8%)	4 (50%)	1 (12.5%)		1 (12.5%)	3 (18.8%)	2 (25%)	1 (12.5%)	1 (12.5%)
**Control-group**	Heifers *n* = 9	1 (11.1%)	1 (11.1%)		1 (11.1%)				2 (22.2%)	2 (22.2%)	2 (22.2%)	1 (11.1%)
	Cows *n* = 5	1 (20%)	2 (40%)	1 (20%)	1 (20%)		1 (20%)		1 (20%)	1 (20%)		

*: in one case the teats appeared filled and were empty the next day; **: in one case the vulva edema became severe to mild the day after.

**Table 2 vetsci-09-00192-t002:** Time (h; median ± median absolute deviation; maximum, minimum) elapsed between first SMS1 (enhanced activity registered over one hour) and SMS2 (high activity continued during the consecutive hour) emitted by the remote calving sensor Moocall^®^ and the onset of calving in heifers (*n* = 16) and cows (*n* = 8; *p* > 0.05).

Animals	First SMS1—Onset of Calving	First SMS2—Onset of Calving
Heifers	21.2 ± 20.2	2.7 ± 3.5
	(max: 95.4; min: 0.1)	(max: 80.6; min: −14.0)
Cows	29.6 ± 29.6	16.4 ± 17.7
	(max: 177.8; min: 0.0)	(max: 176.8; min: −1.7)

**Table 3 vetsci-09-00192-t003:** Percentages of deviations between model predicted birth times and true birth times in the range of half a day (+/− 6 h) or full day (+/− 12 h), plus 95% confidence intervals.

Model	Deviations Smaller +/− 6 h	Deviations Smaller +/− 12 h
SMS1	34.7% 95%-CI: [16%, 57%]	39.1% 95%-CI: [20%, 61%]
SMS2	52.2% 95%-CI: [31%, 73%]	56.5% 95%-CI: [34%, 77%]
Model 1	21.7% 95%-CI: [7%, 44%]	39.1% 95%-CI: [20%, 61%]
Model 2	26.1% 95%-CI: [10%, 48%]	30.4% 95%-CI: [13%, 53%]
Model 3	21.7% 95%-CI: [7%, 44%]	34.8% 95%-CI: [16%, 57%]
Model 4	13% 95%-CI: [3%, 34%]	34.8% 95%-CI: [16%, 57%]

## Data Availability

Not applicable.

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
