# Peer review of "Comparison between a Calving Predictive System and a Routine Prepartal Examination in German Holstein Heifers and Cows"

_vetsci, 2022, doi:10.3390/vetsci9040192_

Round 1

Reviewer 1 Report

The manuscript from Martin et al. is an interesting, although not very novel, study regarding the use of calving devices in dairy cattle. The thematic is very modern, as these devices are becoming increasingly popular among farmers and as such, any new aspect may benefit a veterinarian regarding its on-farm application. The study design is generally unproblematic (although I would personally have preferred to use both moocall and clinical assessment in the same animals) and the study is nicely written. Unfortunately, I have major objections regarding the statistical analysis used by the authors which make the present study unsuitable for publication in its present form. In my opinion, the use of correlation is not correct for this kind of data. The reason is that correlation does not reflect if a set of data is identical, but only if they correlate with its other. For instance, in your data set, if in a hypothetical scenario insemination to first SMS was for every cow constantly 2 days earlier from calving, that would result in a perfect correlation (R=1), but with a zero sensitivity regarding calving prediction within 12 or 24 hours (which is actually the aim when using calving devices). Your analysis should in my opinion be based on the excellent method of Bland-Altman to measure method agreement (i.e., to compare moocall with clinical assessment) and on sensitivity/ specificity / PPV /NPV analysis for their suitability to predict calving within 12 and/or within 24 hours. I think the fact that all studies you cite in your abstract report such values is not random. However, I think a re-analysis of your data will probably change many of your results and conclusions, so that a new revision has to take place in case you decide to do so.  

Author Response

Dear reviewer,
we are very grateful for your suggestions which will enhance the quality of our work. Please, find attached the last version of the manuscript for better understanding of the modifications.

We agree that correlation analysis alone does not sufficiently describe the performance of the models. Therefore, we included residual plots to illustrate the deviations between true and predicted birth times. Residual plots are a standard tool to study performance and justification of assumptions of regression models. Bland-Altman plots are much similar but more suited to compare agreement between two analytical assays. We thank the reviewer for his proposal to use accuracy, sensitivity, specificity and predictive values, but these measures are more suited for dichotomous classifiers but nor for models with metric endpoints. Therefore, we calculated only the proportion of animals for which deviations where in the range of half a day (+/- 6 hours) or of a whole day (+/- 12 hours). We admit that these proportions are rather small. However, our main conclusions remain true, that is SMS alerts can improve calving time predictions, and in particular, a combined model with SMS class 2 alerts and additional traits as predictors performs better than SMS alerts alone or than models without SMS alerts.

We kindly ask you to reconsider the evaluation of our manuscript.

Best regards,

Lara Górriz Martín

Reviewer 2 Report

Dear authors,

with interest I have read your manuscript and I think it is wellworth to be published, it needs some improvement before I can advise the Editor to accept it for publication.

Abstract 

you use a lot of abreviations (e.g. EV, TF and VS) that are not used more in this parrt and can be skipped

p-values max 2 figures behind the comma with this number of data

at the end of the introduction, I miss a clear objective of this study

line 70: you mention here Moocall for the first time and the reader doesn't know what this is. I think you need to add some extra information here 

line 70-71: Is the information about milking necessary here

calving management: I miss information about type of pen and ration

2.3 It is unclear who performed the clinical examination and some parameters are subjective. Have you checked agreement within investigators

line 106: Figure 1 is missing!!

line 133: Gestation length was .. days in heifers and .. days in cows..

line 141: replace findings by outcomes

line 145: you mentioned sectio caesarian but what about fototomie??

line 149: see 133

line 146: and another one...

line 161: see 146

line 162: dead at calving??

line 196: Fig. 3??

Model Presentation: see remark made by Abstract

line 236: was responsible for--> may have influenced

line 238: fig 3. or Model 3

line 253: In this study...

line 263: agrees with the dat of Grunert (1993; 19)

line 278: data--> dataset

line 279: see 238

line 281: to our knowledge ... parameters should be replaced to the start of the discussion

line 294-295: what about the use of camera's

Author Response

Dear reviewer,
we are very grateful for your suggestions, which will enhance the quality of our work. Please, find attached the last version of the manuscript for better understanding of the modifications. Furthermore, our answers to each of your remarks are summarized in the table below.

Best regards,

Lara Górriz Martín

Reviewer remark Authors answer
you use a lot of abreviations (e.g. EV, TF and VS) that are not used more in this parrt and can be skipped All the acronyms referring to the obstetrical traits were removed from the manuscript; only complete words referring to the obstetrical traits are used in the study.
p-values max 2 figures behind the comma with this number of data To answer this remark a citation of New England Journal of Medicine (NEJM) is used: The NEJM states: "Except when one-sided tests are required by study design, such as in noninferiority trials, all reported P values should be two-sided. In general, P values larger than 0.01 should be reported to two decimal places, those between 0.01 and 0.001 to three decimal places; P values smaller than 0.001 should be reported as P<0.001. Notable exceptions to this policy include P values arising in the application of stopping rules to the analysis of clinical trials and genetic-screening studies."
at the end of the introduction, I miss a clear objective of this study The sentence "Therefore, the aims of this study were to investigate the efficiency of Moocall in predicting labor in dairy cows and heifers and to compare calving detection through Moocall with a routine clinical examination at a dairy cattle farm." was added 
line 70: you mention here Moocall for the first time and the reader doesn't know what this is. I think you need to add some extra information here  The sentence "Moocall® (Moocall, Moocall Ltd., Dublin, Ireland) is a device based on an accel-erometer system detecting tail raising and overactivity. Placed at the tail 4 days prior to the predicted calving date (pcd), the Moocall sensor produces alarms in form of short message services (SMS) in case of overactivity." was added.
line 70-71: Is the information about milking necessary here The milking times were removed.
calving management: I miss information about type of pen and ration Added: "All heifers and cows were housed in similar prepartum pens and moved into adjacent individual maternity pens for calving with straw bedding. The animals were moved into the calving pen after first signs of udder filling. The diets of the animals were formulated to meet the nutritional requirements of  dry dairy cows stated by the German Society of Nutrition Physiology (Society of Nutrition Physiology, 2001)."
2.3 It is unclear who performed the clinical examination and some parameters are subjective. Have you checked agreement within investigators The sentences "Independent of the on-farm personnel, a trained veterinarian performed an examination of the animals every 6 hours. The examined traits were changes in pelvic ligament relaxation, edema of the vulva, teat filling, vaginal secretion, flexibility of the tail tip, tail raising and behavior." and "The trained veterinarian did not influence the obstetrical decisions taken by the on-farm personnel." were added
line 106: Figure 1 is missing!! Authors mistake: Figure 1 signed as Figure 2. Mistake was corrected!

line 133: Gestation length was .. days in heifers and .. days in cows.. Corrected: "Gestation length was 276.6 ± 9.1 days in heifers (n=16) and 283.9 ± 4.2 days in cows (n=8)"
line 141: replace findings by outcomes Corrected 
line 145: you mentioned sectio caesarian but what about fototomie?? Corrected "Caesarean sections and fetotomies were not needed in any of the cases"
line 149: see 133 Corrected "Gestation length was 276.5 ± 4.1 days in heifers and 279.2 ± 6.7 days in cows" 
line 146: and another one... Corrected: "Gestation length was 276.5 ± 4.1 days in heifers and 279.2 ± 6.7 days in cows, which did not show a statistically significant difference (mean ± SD)" 
line 161: see 146 Corrected: "Calving monitoring sessions in 6 hours intervals was 10.5 ± 3 (Median ± MAD) in four heifers." 
line 162: dead at calving?? Corrected: "Seven calves from heifers were vital, one was weak at birth and in one further case data were not available. Four calves from cows were vital and one was dead at calving." 
line 196: Fig. 3?? Table 2 is correct
Model Presentation: see remark made by Abstract All the acronyms referring to the obstetrical traits were removed from the manuscript; only complete words referring to the obstetrical traits are used in the study.
line 236: was responsible for--> may have influenced Corrected: "We do not preclude the possibility that Moocall may have influence this observation, however, the fact that 22.2% of the heifers in the control group also showed tail raising, makes it improbable." 
line 238: fig 3. or Model 3 Corrected: "Using likelihood ratio tests, the full model with SMS information (model 3 shown in Fig. 3) showed a significantly " 
line 253: In this study... Corrected "In this study, pelvic ligament relaxation combined…"
line 263: agrees with the dat of Grunert (1993; 19) Corrected: "However, 57% of our animals showed a gestation length shorter than 280 days, which agrees with the data of Grunert (1193; [19]). "
line 278: data--> dataset Corrected: " if the whole dataset is considered, we observed that models without Moocall infor-mation " 
line 279: see 238 See comment to 238
line 281: to our knowledge ... parameters should be replaced to the start of the discussion Comma removed
line 294-295: what about the use of camera's Is a complementary use of cameras and  Moocall  or only cameras meant? A complementary use of Moocall and cameras should be investigated in further studies. Would it not be in any case a clinical examination much more reliable and not so time-consuming?

Reviewer 3 Report

Dear authors:
It is a very interesting paper, well designed and well analysed that offers very technical and
practical information about the examined product.

Typos

Line 11: delete the final “;

Line 15: days instead of day

Line 75, 91, 102, 114,131, 172, 194, 207: delete the spaces, . and /” that appears between
the number of the subchapter and the name of the titles so that it appears uniform
throughout the paper

Line 96: personnel instead of personell

Lines 170-171: Table 1 is misconfigured; it is necessary to correct it so that it can be
understood

Corrections, suggestions, and clarifications

The acronym TE (tail elevation) appears in lines 17 and 98; and the acronym TR (tail raising)
appears in lines 94-95, 240 and 243. For a better understanding it would be convenient to
unify the term.

Lines 82-83: clarify when, or with what clinical signs, cows and heifers were moved to
individual maternity pens for calving

Author Response

Dear reviewer,

we are very grateful for your suggestions, which will improve the quality of our work. Our answers to each of your remarks are summarized in the table below. Furthermore, for the better understanding of the modifications the last version of the manuscript is attached.

Best regards,

Lara Górriz Martín

Reviewers Remark Authors Answer
Line 96: personnel instead of personell corrected
Lines 170-171: Table 1 is misconfigured; it is necessary to correct it so that it can be understood corrected; please, let me know, if the new configuration is ok.
The acronym TE (tail elevation) appears in lines 17 and 98; and the acronym TR (tail raising) appears in lines 94-95, 240 and 243. For a better understanding it would be convenient to unify the term. All the acronyms referring to the obstetrical traits were removed from the manuscript; only complete words referring to the obstetrical traits are used in the study
Lines 82-83: clarify when, or with what clinical signs, cows and heifers were moved to individual maternity pens for calving Added: "All heifers and cows were housed in similar prepartum pens and moved into adjacent individual maternity pens for calving with straw bedding. The animals were moved into the calving pen after first signs of udder filling. The diets of the animals were formulated to meet the nutritional requirements of  dry dairy cows stated by the German Society of Nutrition Physiology (Society of Nutrition Physiology, 2001)."

Round 2

Reviewer 1 Report

The manuscript has much improved with the changes the authors have performed. Although, I am still not fully satisfied with the analysis, the reader can now better understand the relative low calving prediction possibilities that both moocall and clinical signs offer. This is in my opinion of paramount importance, because it is now more clear that farmer's cannot fully rely on moocall and calving surveillance cannot be substituted to 100 %. Additionally, veterinarians can derive data from residual plots to have a better estimation regarding calving time from moocall alerts.  

Author Response

Dear Reviewer,

thank you very much for your comments and for this kind of cooperation! It all definetly contributed to improve the quality of the work!

English language and style fine/minor spell check is planed. 

Best regards,

Lara Górriz Martín

Reviewer 2 Report

Dear authors,

Although it is interesting< I still have serious problems with the presentation of your p-values. With this no. of animals you only can present this with 2 figures in abstract and results (incl. tables) behind the dot. this presantion with the variation of 2 3 figures is very disruptive and not acceptable for me.

line 97: 1 vet or more was this checked for agreement 

Author Response

We agree with the reviewer and have harmonized the presentation of the p-values.